# Van der Waals epitaxial growth and optoelectronics of large-scale WSe$_2$/SnS$_2$ vertical bilayer p–n junctions

Tiefeng Yang[1], Biyuan Zheng[1], Zhen Wang[2], Tao Xu [3], Chen Pan[4], Juan Zou[1], Xuehong Zhang[1], Zhaoyang Qi[1], Hongjun Liu [1], Yexin Feng[1], Weida Hu[2], Feng Miao [4], Litao Sun [3], Xiangfeng Duan[5] & Anlian Pan[1]

High-quality two-dimensional atomic layered p–n heterostructures are essential for high-performance integrated optoelectronics. The studies to date have been largely limited to exfoliated and restacked flakes, and the controlled growth of such heterostructures remains a significant challenge. Here we report the direct van der Waals epitaxial growth of large-scale WSe$_2$/SnS$_2$ vertical bilayer p–n junctions on SiO$_2$/Si substrates, with the lateral sizes reaching up to millimeter scale. Multi-electrode field-effect transistors have been integrated on a single heterostructure bilayer. Electrical transport measurements indicate that the field-effect transistors of the junction show an ultra-low off-state leakage current of $10^{-14}$ A and a highest on–off ratio of up to $10^7$. Optoelectronic characterizations show prominent photo-response, with a fast response time of 500 μs, faster than all the directly grown vertical 2D heterostructures. The direct growth of high-quality van der Waals junctions marks an important step toward high-performance integrated optoelectronic devices and systems.

[1] Key Laboratory for Micro-Nano Physics and Technology of Hunan Province, State Key Laboratory of Chemo/Biosensing and Chemometrics, and School of Physics and Electronics, Hunan University, Changsha, 410082 Hunan, China. [2] State Key Laboratory of Infrared Physics, Shanghai Institute of Technical Physics, Chinese Academy of Sciences, 200083 Shanghai, China. [3] SEU-FEI Nano-Pico Center, Key Lab of MEMS of Ministry of Education, Southeast University, 210096 Nanjing, China. [4] National Laboratory of Solid State Microstructures, School of Physics, Collaborative Innovation Center of Advanced Microstructures, Nanjing University, 210093 Nanjing, China. [5] Department of Chemistry and Biochemistry and California NanoSystems Institute, University of California at Los Angeles, Los Angeles, CA 90095, USA. Tiefeng Yang and Biyuan Zheng contributed equally to this work. Correspondence and requests for materials should be addressed to W.H. (email: wdhu@mail.sitp.ac.cn) or to A.P. (email: anlian.pan@hnu.edu.cn)

The emerging two-dimensional (2D)-layered semiconductors have shown considerable potential for designing next-generation integrated electronic and optoelectronic systems, due to their many unique physical and structural properties[1–14]. In particularly, diverse 2D-layered semiconductors can be flexibly combined to form diverse vertical van der Waals (vdWs) heterostructures[15–23] with atomically sharp interfaces and tunable band alignment, opening up vast opportunities for fundamental investigation of novel electronic and optical properties at the limit of single atom thickness and potential applications in novel device concepts[9, 17, 19, 21, 24–30]. The current studies of 2D van der Waals heterostructures have been largely limited to the mechanically exfoliated and restacked flakes[17, 20, 26, 31–34], which is arduous and clearly un-scalable for practical technologies. Some recent progresses have also shown that 2D vertical heterostructures can be produced by direct vapor phase growth[35–42]. Compared to the mechanical stacking approach, the direct growth strategy could offer unique advantages of easy size control, clean interface, and potential for practical industrial applications. However, direct vapor growth of large-scale high-quality 2D atomic layered vertical heterostructures, especially p–n junctions, remains a great challenge.

As an important p-type 2D semiconductor with excellent physical properties, layered tungsten diselenide (WSe$_2$) has attracted great attention as a promising material for future scaled device applications[43]. Tin disulfide (SnS$_2$), as a member of IV–VI A group, and being an important n-type layered semiconductor, has drawn considerable attention due to the advantages of low-cost, earth-abundant, nontoxic, and enviromentally friendly[34, 44, 45]. Recently, 2D heterostructures combining the p-type WSe$_2$ and n-type SnS$_2$ have aroused great interest. Wang et al.[34] have reported the preparation of few-layer/few-layer stacked WSe$_2$/SnS$_2$ device through a mechanically exfoliated and restacked method, and investigated their anti-ambipolar behavior. Zhang et al.[35] demonstrated the growth of few layers of WSe$_2$ on the pre-prepared randomly oriented micaoplates of SnS$_2$. Herein, we report a two-step vapor phase route to controlled growth of large-scale WSe$_2$/SnS$_2$ vertical bilayer p–n junctions on SiO$_2$/Si. The as-

grown junctions are highly crystallized, with their lateral sizes reaching up to millimeter scale, representing the largest size of atomic layered vertical heterostructures ever been achieved. Backgate field-effect transistors were fabricated with high on–off ratio, ultra-low leakage current, and show fast photoresponse speed comparing favorably to mechanically staked 2D vertical junctions. The direct growth of high-quality van der Waals junctions marks an important step toward high-performance integrated optoelectronic device and systems.

## Results

**Characteristics of WSe$_2$/SnS$_2$ heterostructure.** Figure 1a shows a top view of atomic structure illustration of the WSe$_2$/SnS$_2$ bilayer heterostructure, in which the bottom WSe$_2$ monolayer and the top SnS$_2$ monolayer are stacked by weak vdWs force. Figure 1b shows the relative band alignment of the bilayer WSe$_2$/SnS$_2$ heterostructure calculated by Vienna ab initio simulation package (VASP). The calculation details could be found in the Supplementary Figs 1 and 2. The valence band maximum of WSe$_2$ is higher than that of the conduction band minimum of SnS$_2$, forming a type-III broken-gap heterojunction[17, 20]. Both of the conduction band offset ($\Delta E_c$, 1.69 eV) and the valence band offset ($\Delta E_v$, 1.64 eV) of WSe$_2$/SnS$_2$ are larger than those of any other reported directly grown 2D heterostructures[36, 37, 40, 41, 46, 47]. The large band offset may cause efficient interlayer charge transfer between the bottom and the top layers, providing an ideal structural base for high-performance optical and electrical applications[40, 48].

The WSe$_2$/SnS$_2$ vertical bilayer heterostructures were grown via a two-step chemical vapor deposition (CVD) strategy. Large-scale WSe$_2$ monolayers were first grown on the SiO$_2$/Si substrate, through a controlled vapor deposition process. The pre-prepared large-scale WSe$_2$ monolayers were then used as templates for subsequent van der Waals epitaxial growth of SnS$_2$ monolayers in vapor phase to achieve the vertical bilayer heterostructures. The two-step vapor growth process is shown schematically in Fig. 1c, and the detailed descriptions about the growth are included in Method section.

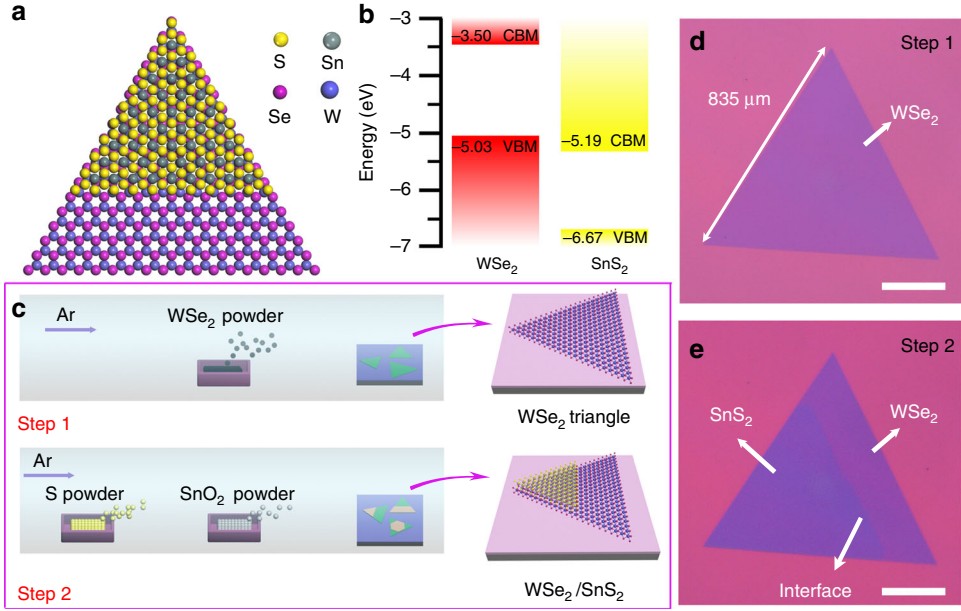

**Fig. 1** Synthesis of large-scale type-III WSe$_2$/SnS$_2$ heterostructure. **a** Schematic of vertically stacked WSe$_2$/SnS$_2$ vdW heterostructure. **b** Band alignment of type-III WSe$_2$/SnS$_2$ junction. **c** Schematic illustrating the two-step vapor expitaxy growth of WSe$_2$/SnS$_2$ heterostructure. **d** Typical optical image of as-grown large-scale monolayer WSe$_2$ triangular flake achieved after step 1. **e** Optical image of as-grown vertically stacked WSe$_2$/SnS$_2$ heterostructure after step 2 by using the same flake in Fig. 1d. All the scale bars: 200 μm

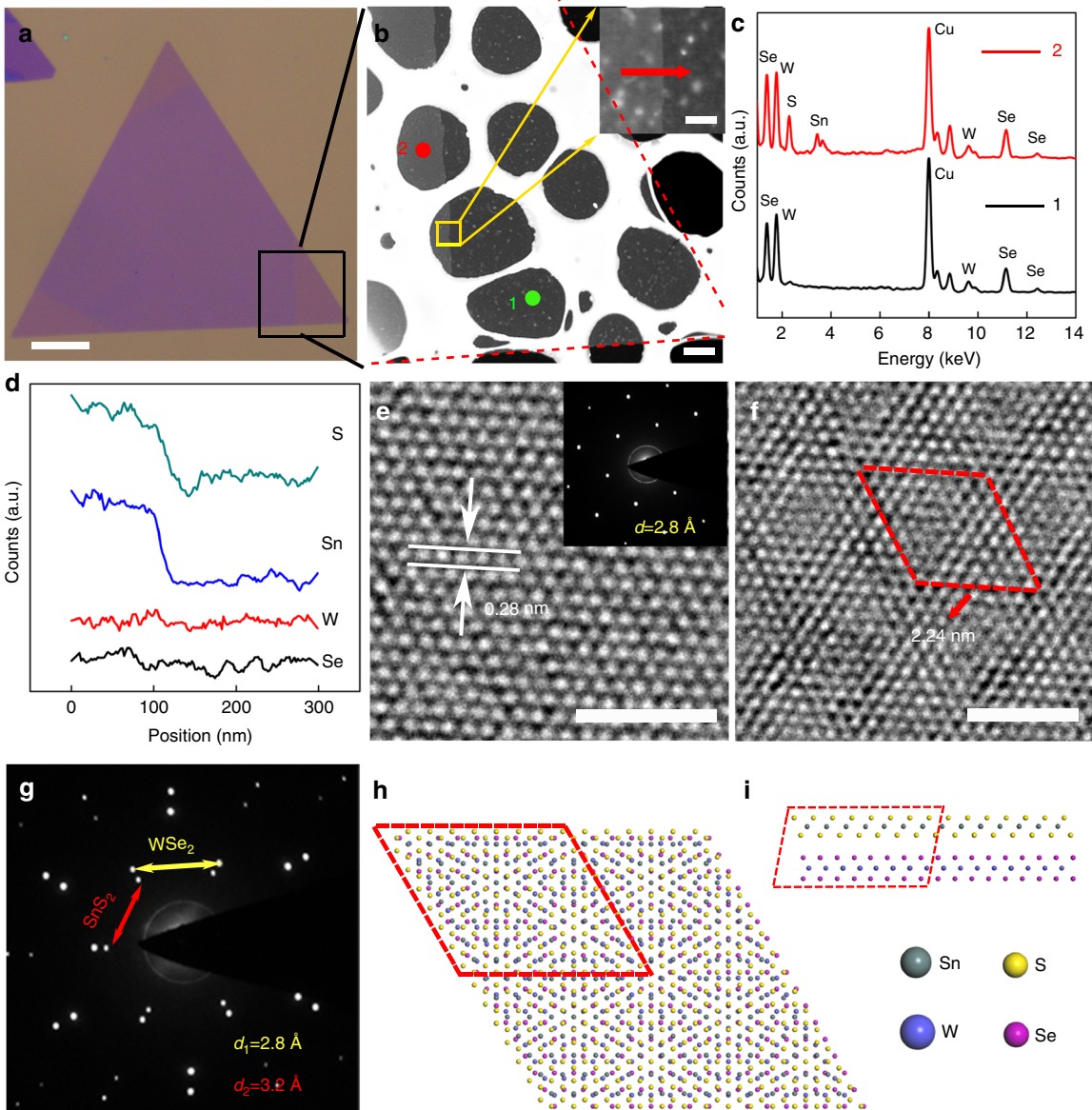

**Fig. 2** Atomic structure of the vertically stacked $WSe_2/SnS_2$ heterostructure. **a** A typical optical image of the heterostructure with three interfaces at the corners of the triangle flake. Scale bar, 30 μm. **b** Low-resolution TEM image of the interface from the black rectangle region in **a**. Scale bar, 5 μm. Inset is the enlarged interface image highlighted in the yellow rectangle. Scale bar, 100 nm. **c** EDS results of point 1 and point 2 in **b**, marked by green and red colors, respectively. **d** Line scans of S, Sn, W, Se elements distribution across the red arrow in the inset of **b**. **e** HRTEM image taken on the green point 1 region in **b**. Scale bar, 2 nm. Inset is the SAED pattern of the bare $WSe_2$. **f** HRTEM image taken on the red point 2 region in **b**, indicating the vertical stacked $WSe_2/SnS_2$ vdW heterostructure with Moiré pattern. Scale bar, 2 nm. **g** The SAED pattern collected from point 2 region shows two sets of electron diffraction patterns, corresponding to $WSe_2$ and $SnS_2$, respectively. **h**, **i** Top view and side view of the atomic model of the $WSe_2/SnS_2$ vdW heterostructure, respectively. The supercell is highlighted by red dashed lines

For further understanding the two-step growth process, we have closely characterized the same $WSe_2$ monolayer domain before and after the second step growth of $SnS_2$ monolayer. Figure 1d gives the optical image of a typical as-grown large-scale $WSe_2$ triangle flake, with the size measured to be 835 μm, and Fig. 1e is the corresponding $WSe_2/SnS_2$ vertical heterostructure obtained after the second step growth. The pre-grown $WSe_2$ triangular domain is partially covered by the $SnS_2$ domain, which can be distinguished from the optical contrast in the image. The size of the vertical junction can be well controlled by the growth time of both the bottom $WSe_2$ and the top $SnS_2$ layers, with the maximum lateral size reaching up to millimeter scale, which is about one order of magnitude larger than both the staked or synthetic vertical bilayer heterostructures reported previously[36, 37, 40–42, 49]. More detailed morphology information can be found in Supplementary Fig. 3.

Transmission electron microscope (TEM) measurements were conducted to further investigate the crystal quality and structure of the as-grown $WSe_2/SnS_2$ heterostructures. Figure 2a shows the optical image of a typical $WSe_2/SnS_2$ heterostructure for TEM characterization and Fig. 2b shows its corresponding TEM image at the marked interface region of the sample after transferred onto a copper grid (see method), with the inset showing the local high-magnification image of the interface. Fig. 2c shows the elemental energy dispersive X-ray spectroscopy (EDS) collected from both the monolayer (green, point 1 in Fig. 2b) and bilayer

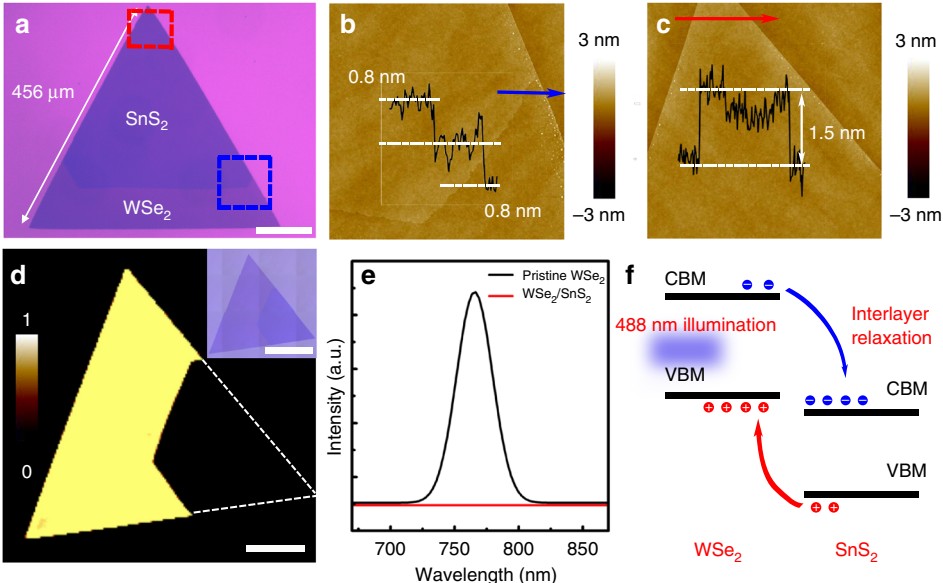

**Fig. 3** AFM and PL characterization of the vertically stacked WSe₂/SnS₂ heterostructure. **a** Optical micrograph of a large-scale 1 L SnS₂/1 L WSe₂ stacked triangular flake grown on SiO₂/Si. Scale bar, 100 μm. **b** AFM image of the blue rectangle region in **a**. Inset is height profile along blue arrow, indicating the thickness of bottom layer and top layer is 0.8 and 0.8 nm, respectively. Corresponding to 1 L SnS₂ on 1 L WSe₂. **c** AFM image of the red rectangle region in **a**. Inset is the height profile across the red arrow, indicating the thickness of the heterostructure is 1.5 nm (1 L WSe₂ + 1 L SnS₂). **d** PL intensity mapping (720–780 nm). Scale bar, 60 μm. Inset image is the optical image of the sample used for PL characterization. Scale bar, 100 μm. **e** PL spectra observed from bare WSe₂ region (black curve) and WSe₂/SnS₂ vdW heterostructure region (red curve). **f** Schematic diagram of WSe₂/SnS₂ heterostructure band structure and photoexcitation, interlayer relaxation process in WSe₂/SnS₂ heterojunction

(red, point 2 in Fig. 2b) regions, respectively. Except the Cu signal from the grid, only elements W and Se were collected from the monolayer region, while apart from W and Se, elements Sn and S were also detected at the bilayer region, indicating the formation of WSe₂/SnS₂ heterostructure, which can also be well demonstrated by the element line scans across the interface region (Fig. 2d). The high resolution transmission electron microscope (HRTEM) image and the selected electron diffraction pattern (SAED) (Fig. 2e) show the bottom region is well crystallized with the measured lattice spacing of 0.28 nm, well consistent with the value of (100) plane spacing of WSe₂. The bilayer heterostructure region shows obvious Moiré patterns (Fig. 2f), caused by the overlapping lattices between SnS₂ and WSe₂. The smallest periodic repeated cell is demarcated by the red dashed rhombus, with a lattice constant of 2.24 nm. The corresponding SAED shows two different sets of hexagonally arranged diffraction patterns (Fig. 2g), with the calculated lattice spacings consistent with the SnS₂ (0.32 nm) and WSe₂ (0.28 nm), respectively[44]. The large lattice misfit (14.3%) indicates that the SnS₂ domain is stacked on the WSe₂ through vdW epitaxy[36, 37, 40–42]. Figure 2h and i gives the top and side views of the theoretical atomic structure model of the WSe₂/SnS₂ heterostructure with Moiré patterns, consistent with the experimental observation shown in Fig. 2f, with the periodic repeated cell marked by the red dashed rhombus, corresponding to 7 × 7 SnS₂ stacked on 8 × 8 WSe₂[41].

The thickness of the as-grown heterostructures was further confirmed by atomic force microscope (AFM) measurements. Figure 3b and c is the AFM images detected at the selected stack region (red dashed rectangle) and the interface region across from SnS₂ to WSe₂ (blue dashed rectangle), respectively, as marked in Fig. 3a. The insets of the AFM images show the corresponding line scan height profile, indicating that the thickness of both the bottom and the top layer are 0.8 nm, with a total thickness at the stack region of 1.5 nm, well demonstrating the achieved bilayer heterostructures (1 L WSe₂ + 1 L SnS₂).

Room temperature photoluminescence (PL) mapping (720 nm - 780 nm) and their corresponding local spectra obtained from a typical heterostructure domain with most of its surface being bilayer stacked (Fig. 3d, e). The results indicate that the bottom uncovered WSe₂ monolayer exhibits strong PL emission with a dominant emission peak locating at 766 nm, corresponding to the recombination of excitons[50, 51], whereas the bilayer WSe₂/SnS₂ region shows apparent PL quenching, with essentially no detectable PL (Fig. 3e). The PL quenching in the stack region indicates the interaction induced energy (charged carrier) transfer between the WSe₂ and the SnS₂[48]. According to the band alignment of WSe₂/SnS₂ as shown in Fig. 1b, the conduction band minimum of SnS₂ is about 1.78 eV lower than that of WSe₂, whereas the valence band maximum of SnS₂ is about 1.67 eV lower than that of WSe₂. Thus, the photo-excited electrons and holes in WSe₂ prefer to transfer to low-energy states in SnS₂, rather than forming excitons in WSe₂, However, being a typical indirect semiconductor, SnS₂ is normally inradiative[44, 52]. The charge transfer from WSe₂ to SnS₂ will consequently lead to the significant PL quenching of the heterostructures (Fig. 3e), as schematically exhibited in Fig. 3f[37, 40, 41, 46, 48, 53, 54].

**Electrical transport properties of WSe₂/SnS₂ heterostructure.** To further investigate the electrical charge transport properties and optoelectronic performance of the resulted WSe₂/SnS₂ vdW heterostructures, multi-electrode backgate field-effect transistors (FETs) were designed and fabricated based on a partially covered heterostructure domain grown at a (P++) Si/SiO₂ substrate, with Ti/Au thin film as the source-drain electrodes and the silicon substrate as the backgate contact electrode, as shown schematically in Fig. 4a. This device structure is suitable for systematically investigating the performance of different devices integrated on the same nanostructure. All the device measurements were conducted under vacuum at room temperature.

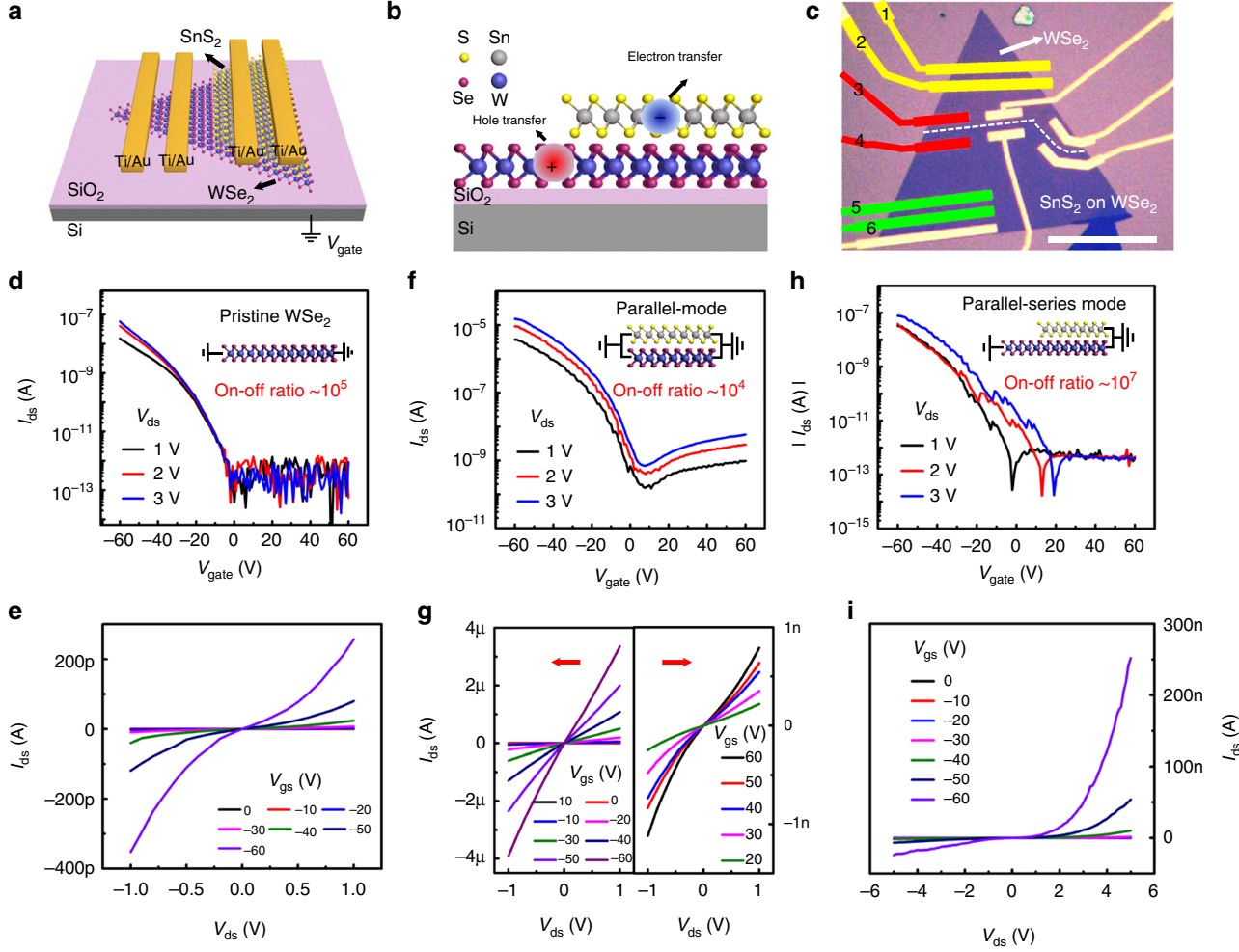

**Fig. 4** Electrical characterization of the vertically stacked WSe$_2$/SnS$_2$ heterostructure. **a** Schematic diagram of the multi-electrode WSe$_2$/SnS$_2$ vdW heterostructure backgate device. **b** Front view of the WSe$_2$/SnS$_2$ vdW heterostructure. **c** An optical image of fabricated WSe$_2$/SnS$_2$ vdW heterostructure device. Scale bar, 60 μm. **d** $I_{ds} - V_{bg}$ curves of the pristine WSe$_2$ measured using electrodes 1 and 2, demonstrating p-type behavior. **e** $I_{ds} - V_{ds}$ output characteristics of pristine WSe$_2$ FET at various backgate voltages. **f** $I_{ds} - V_{bg}$ curves of the parallel-mode vdW heterostructure measured using electrodes 5 and 6, demonstrating ambipolar behavior. **g** $I_{ds} - V_{ds}$ output characteristics of the parallel-mode vdW heterostructure at various backgate voltages from −60 V to 10 V and from 20 V to 60 V. **h** $I_{ds} - V_{bg}$ curves across the p (WSe$_2$) − n (SnS$_2$) heterojunction, named parallel-series mode, demonstrating p-type behavior (measured using electrodes 3 and 4). **i** $I_{ds} - V_{ds}$ output characteristics across the p–n junction at different backgate voltages

Figure 4b gives schematically the front view of the heterostructure, with electron dominating the charge transfer process in the n-type SnS$_2$ layer and hole dominating transfer in the p-type WSe$_2$ layer, as reported in the literatures[44, 55–57]. The optical image of a typical WSe$_2$/SnS$_2$ vdW heterostructure after device fabrication is shown in Fig. 4c. Electrodes 1, 2, and 3 were deposited on top of the pristine WSe$_2$ region, whereas electrodes 4, 5, and 6 were deposited on the top of the stack WSe$_2$/SnS$_2$ region. As a result, three kinds of device channels were formed, (1) pristine WSe$_2$ channel (electrodes 1 and 2, yellow), (2) parallel-mode channel (electrodes 5 and 6, green, bilayer stacked region) and (3) parallel-series mode channel (electrodes 3 and 4, red, heterojunction region), respectively. The transport characteristic of pristine WSe$_2$ shows typical p-type behavior, indicating that holes dominate charge transport process, with an on–off ratio of $10^5$ and a threshold voltage of −2 V (Fig. 4d). The corresponding output characteristic in Fig. 4e shows that the $I_{ds}$ decreases with the $V_g$ varying from −60 to 0 V, and the $I_{ds} - V_{ds}$ curves shows an inflection point at near $V_{ds} = 0$ V, indicating the existence of a Schottky barrier at the WSe$_2$ contacting with Ti/Au. The field-effect charge carrier mobility

is calculated to be 0.02 cm$^2$ V$^{-1}$ s$^{-1}$ ($L = 5.2$ μm, $W = 36.5$ μm, $V_{ds} = 1$ V). In addition, the transport characteristic of the parallel-mode device shows ambipolar behavior (Fig. 4f), which is attributed to the p-type WSe$_2$ and the n-type SnS$_2$[41]. The corresponding output characteristic show that the $I_{ds} - V_{ds}$ have very good linear relation at negative $V_g$ values, and $I_{ds}$ decreases as the back voltage varies from −60 to 10 V, following an increase as $V_g$ varies from 10 to 60 V (Fig. 4g). The hole mobility is calculated to be 10.1 cm$^2$ V$^{-1}$ s$^{-1}$ ($L = 4.25$ μm, $W = 51$ μm, $V_{ds} = 1$ V), which is three orders higher than that of the pristine WSe$_2$. Moreover, the device shows an on–off ratio of $10^4$ with a saturation current reaching up to 10 μA ($V_{ds} = 1$ V). The electric transport properties of the parallel-series mode device were measured using electrodes 3 and 4. The transport characteristic is p-type dominant, indicating that holes in the bottom WSe$_2$ dominate the charge transport across this junction. The hole mobility is calculated to be 0.149 cm$^2$ V$^{-1}$ s$^{-1}$ (Fig. 4h) ($L = 9.35$ μm, $W = 21.2$ μm, $V_{ds} = 1$ V), one order higher than that of the pristine WSe$_2$ device. Importantly, benefit from the existence of the barrier near the vertical junction, leading to a very low leak current ($10^{-14}$ A), and an ultrahigh on–off ratio to $10^7$, two

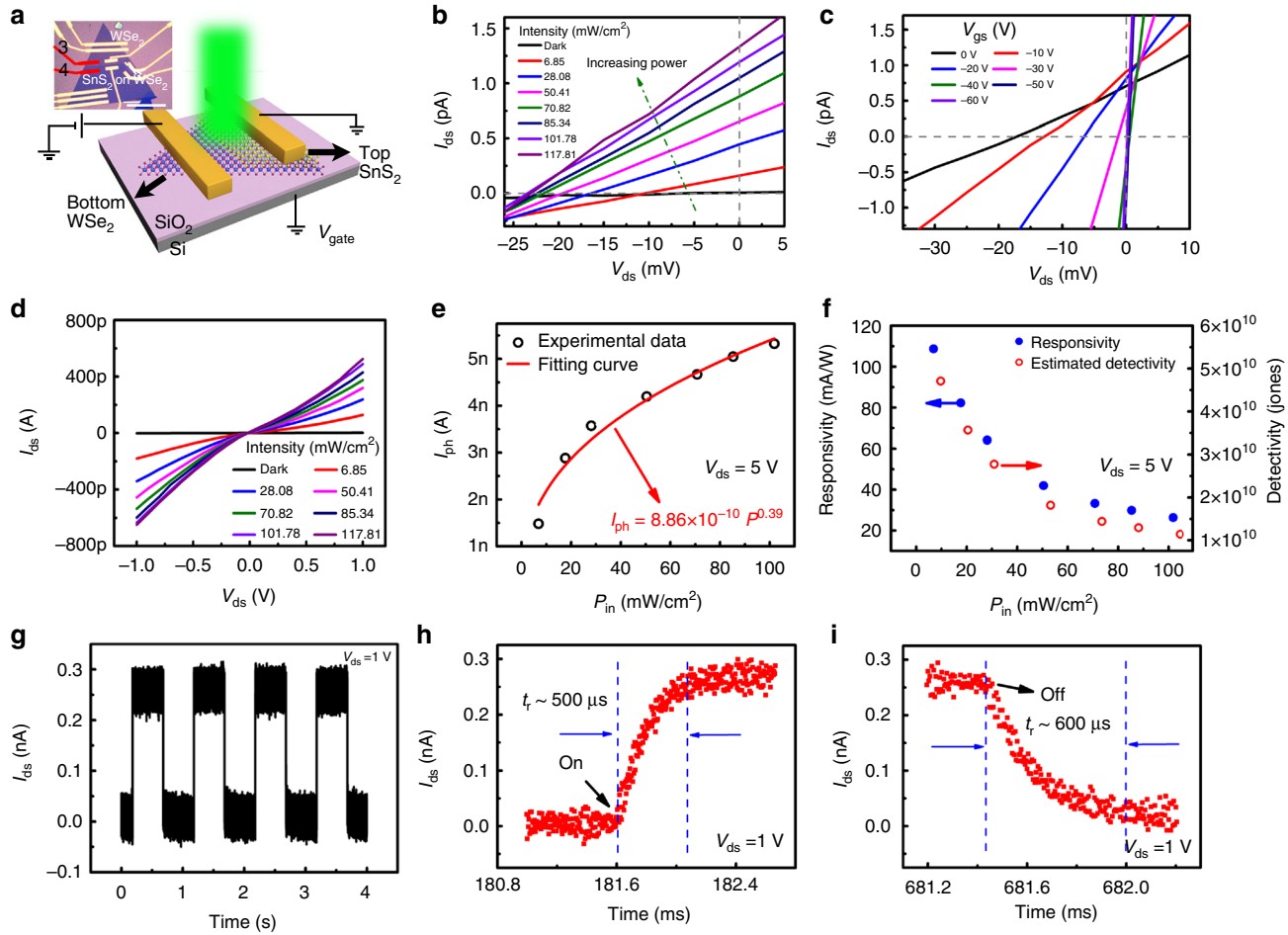

**Fig. 5** Optoelectronic characterization of the parallel-series mode WSe$_2$/SnS$_2$ heterojunction. **a** A cartoon schematic of the parallel-series mode WSe$_2$/SnS$_2$ vdW heterojunction photodetector. Inset is an optical image of the heterojunction device. (measured using electrodes 3 and 4, see Fig. 4c). **b** $I_{ds}-V_{ds}$ curves across the p–n junction under 520 nm laser illumination at different incident power. **c** $I_{ds}-V_{ds}$ curves at various backgate voltages from −60 V to 0 V under 520 nm laser illumination at fixed 101 mW cm$^{-2}$ power intensity. **d** $I_{ds}-V_{ds}$ curves across the p–n junction under 520 nm laser illumination at different incident power. $V_{ds}$ range from −1 V to 1 V. **e** Dependence of photocurrent on illumination power intensities. **f** Photoresponsivity and detectivity of the photodetector at various illumination power intensities. **g** Photocurrent response of the device, the laser light is turned on/off by a chopper worked at 1 Hz (520 nm, 101 mW cm$^{-2}$, $V_{ds} = 1$ V). **h**, **i** Time-resolved photoresponse of the device, namely the rise and fall times of the photocurrent measured at $V_{ds} = 1$ V

orders higher than that of the pristine WSe$_2$ device ($10^5$) and three orders higher than that of the parallel-mode device ($10^4$). Gate-tunable output curves show obvious current rectification behavior (Fig. 4i), indicating a p–n diode is formed across the heterojunction.

It is well reported that vapor grown TMDs monolayers usually have very low mobility, mainly caused from the carrier scattering by the inevitable lattice defects induced during the vapor synthesis and device fabrication processes[58, 59], which is also demonstrated in the pristine WSe$_2$ monolayer device. The mobility can be greatly improved from the heterostructured channels, i.e., the parallel-mode and the parallel-series mode devices, which can be attributed to several reasons. Firstly, the electrons and holes can be efficiently separated with less interactive scattering and faster transfer in the type-III band alignment heterostructures[40]. Secondly, the van der Waals force in the interface will also help to suppress the extrinsic interfacial impurities to increase the mobility[60, 61]. Thirdly, comparing to a single layer of WSe$_2$, the increased carrier density in the heterostructure will enhance the screening of the interfacial Coulomb potential, which will also help to improve the mobility[61].

**Photovoltaic and photoresponse properties of WSe$_2$/SnS$_2$ heterostructure**. The photovoltaic and photoresponse properties of the WSe$_2$/SnS$_2$ p–n heterojunction (parallel-series mode) were investigated using electrodes 3 and 4 (see Fig. 4c). All the measurements were conducted by applying a drain voltage $V_{ds}$ on WSe$_2$ (p-type, terminal "d"), with SnS$_2$ (n-type, terminal "s") being grounded (Fig. 5a).

Furthermore, obvious photovoltaic effect was observed in the direct grown WSe$_2$/SnS$_2$ p–n heterojunction when the junction was under 520 nm laser illumination, which further demonstrate the formation of the atomic junction. As shown in Fig. 5b, with increasing the laser power, more photoinduced electro-hole pairs will separate and contribute to a higher $I_{sc}$. Meanwhile, more electrons and holes excited by the laser illumination aggregate in the interface region and enhance the band bending, which will further increase the absolute value of $V_{oc}$. In addition, as shown in Fig. 5c, when the backgate voltage varies from 0 to −60 V, the absolute value of $V_{oc}$ shows a decrease, with $I_{sc}$ changed slightly. The Fermi level of WSe$_2$ moved down under negative gate voltages, leading to a less band bending and decrease the built-in potential difference, result in a smaller $V_{oc}$. At the same time, both negative backgate voltage and the laser illumination will

trigger an excess carrier concentration, which will reduce the lifetime of carrier and simultaneously increase the recombination. As a result, the $I_{sc}$ changes slightly[17, 41].

Photoconductive properties of the parallel-series mode device under different laser power were also examined, as shown in Fig. 5d. $I_{ph}$ is the photocurrent which is defined as $I_{ph} = I_{light} - I_{dark}$, and the obtained data under different laser power are shown in Fig. 5e, fitted with an equation of $I_{ph} = aP^\alpha$. In our experiments, the fitted parameters of $a$ and $\alpha$ are $8.86 \times 10^{-10}$ and 0.39, respectively. The responsivity ($R$) is calculated by the equation of $R = I_{ph}/PA$, where $I_{ph}$ is the photocurrent, $P$ is the incident light power density, and $A$ is the effective area of the device channel. The responsivity of the device can reach up to 108.7 mA W$^{-1}$ ($P$ = 13.63 nW, $V_{ds}$ = 5 V, device area of 198.2 μm$^2$), which is a lot higher than the ever repoted 2D vapor epitaxy grown p–n junction[41]. Detectivity ($D^\star$) is used to characterize the sensitivity of a photodetector. Assuming that shot noise from the dark current is the major factor limiting the $D^\star$, which can be estimated by $D^\star = RA^{1/2}/(2e\,I_{dark})^{1/2}$, where $R$ is responsivity, $A$ is the effective area of the device channel, $e$ is the electronic charge, and $I_{dark}$ represents the dark current[62, 63]. The maximum estimated $D^\star$ value of $4.71 \times 10^{10}$ Jones can be obtained based on this expression (Fig. 5f). To further investigate the response speed of the WSe$_2$/SnS$_2$ p–n heterojunction photodetector, time-resolved photoresponse measurements were performed by turning on and off the laser light (520 nm, 100 mW cm$^{-2}$) with a chopper worked at 1 Hz. A high-speed oscilloscope was used to monitor the fast-varying signal. As shown in Fig. 5g, the photodetector exhibits the excellent stability and reliability with the on/off photoswitching behavior at $V_{ds}$ = 1 V without additional gate voltage. The rise time ($\tau_r$) and the fall time ($\tau_f$) is 500 μs and 600 μs, respectively, as shown in Fig. 5h, i. Importantly, as far as we known, the response speed of the WSe$_2$/SnS$_2$ heterojunction photodetector is faster than all the CVD directly grown vertical stacked 2D heterostructures[37, 40–42, 64], even two orders of magnitude faster than most mechanical exfoliated and restacked vertical 2D heterostructures[65–67], and nearly four orders of magnitude faster than the response time of the reported pristine WSe$_2$ detector[68, 69].

To further systematically understand the high-performance of the junction (parallel-series mode) detector, the photodetectors of the pristine WSe$_2$ monolayer and the parallel-mode vertical stacked structure based on the same flake were also fabricated and comparatively investigated, as shown in Supplementary Figs. 4 and 5. From the results, the performances of the heterostructured devices (parallel-mode and parallel-series mode) are overall a lot higher than those of the pristine device. For the parallel-series mode devices, the photoinduced electron-hole pairs can be efficiently separated due to the large band offset (Fig. 1b). Meanwhile, benefit from the higher mobility comparing to pristine WSe$_2$, the electrons and holes can be transported faster to the opposite terminals after separation. Both these two factors help increasing the $R$ value and the response speed. However, having a large dark current ($10^{-8}$ A), the photodetectivity is greatly limited for the parallel-mode device, and this disadvantages can be greatly improved in the parallel-series mode device, which possesses the advantages of both efficiently separation, fast charge transfer and also maintain low dark current ($10^{-12}$ A at $V_{ds}$ = 1 V). As a result, the achieved 2D large-scale WSe$_2$/SnS$_2$ heterojunction photodetectors show great potential for high-speed and weak signal detection applications in integrated optoelectronic applications.

## Discussion

We have demonstrated direct van der Waals epitaxial growth of vertical bilayer WSe$_2$/SnS$_2$ p–n junction. The as-grown junctions are highly crystallized, with their lateral sizes reaching up to millimeter scale, representing the largest size of atomic layered vertical heterostructures ever been achieved. TEM results have shown obvious periodic Moiré patterns, indicating a large lattice mismatch (14.3%), further demonstrating the SnS$_2$ domain is stacked on the WSe$_2$ through vdW epitaxy. Strong PL quenching in the bilayer region of WSe$_2$/SnS$_2$ heterostructure was observed, coming from the efficient interlayer charge transfer between the bottom and the top layers. Multi-electrodes backgate FETs were constructed, result in three different kinds of devices integrated on one typical heterostructure flake. Their perfomance of FETs and photodetectors were systematically investigated, indicating the parallel-series mode WSe$_2$/SnS$_2$ p–n junction exhibits an ultra-low leak-off current ($10^{-14}$ A), and a highest on–off ratio ($10^7$). Devices based on the parallel-series mode WSe$_2$/SnS$_2$ p–n junction exhibit an obvious positive promotion in photo-responsvity (108.7 mA W$^{-1}$), photodetectivity ($4.71 \times 10^{10}$ Jones) and photoresponse speed (500 μs), comparing to the pristine WSe$_2$, with all the values improved than all the ever reported direct grown 2D vertical p–n junctions. This study of WSe$_2$/SnS$_2$ van der Waals heterostructures marks a important step toward high-performance integrated optoelectronic devices and systems.

## Methods

**Materials synthesis**. For the first growth of WSe$_2$ monolayer, tungsten diselenide powder was placed at the center of furnace, and a piece of SiO$_2$/Si substrate was placed at the downstream of the quartz tube. At the beginning, 400 SCCM Ar was flowed into the tube for 15 min to ensure a stable chemical reaction environment. Then the flow rate of Ar was controlled at 50 SCCM and the center temperature of the furnace was heated to 1100 °C, keeping at this temperature for ten minutes. For the second growth, three quartz boats loaded with S powder, SnO$_2$ powder, and a piece of SiO$_2$/Si substrate with as-grown WSe$_2$ monolayers were placed at the upstream, center and downstream of the quartz tube, respectively. After the air inside the tube had been purged by Ar flow, the furnace was heated to 600 °C and kept for 8 min. During the growth process, the Ar flow was 50 SCCM and the pressure inside was 8 Torr. After grwoth, the furnace was cooled down to room temperature naturely.

**Characterizations of as-grown WSe$_2$/SnS$_2$ heterostructures**. The morphologies of WSe$_2$/SnS$_2$ heterostructures were characterized using optical microscopy (Zeiss Axio Scope A1), AFM (Bruker Multimode 8). PL measurements were conducted by using a confocal microscope (WITec, alpha-300) with an objective focused 488 nm laser. For the TEM characterization of the sample, the nanosheets were transferred onto grid of copper using a PMMA-assisted positioning transfer method. SiO$_2$/Si wafer accompanied with WSe$_2$/SnS$_2$ heterostructures was coated with PMMA (950 K, A3) by spin-coating at a speed of 2000 RPM for 1 min, then baked the wafer at 180 °C for 2 h. After that, the target WSe$_2$/SnS$_2$ heterostructure flake was located by using the coordinates with an optical microscope. Then the edge of the baked wafer was round up with scotch tape, and subsequently immersed into the KOH (15 M) solution for 12 h. Then the PMMA film was taken out from the KOH solution and swilled fully with DI water. The cleaned PMMA film was removed onto a grid of copper and leaved the grid in the atmosphere of acetone vapor at 40 °C. Finally, the PMMA film was taken away by acetone vapor, leaving the target WSe$_2$/SnS$_2$ heterostructure on the grid of copper.

**Fabrication and measurement of the as-grown WSe$_2$/SnS$_2$ devices**. First, a layer of MMA copolymer (EL6, Microchem Company) was spin-coated on the SiO$_2$ (300 nm)/Si substrate with WSe$_2$/SnS$_2$ heterostructures, followed by a 1 min bake at 160 °C on the hot plate. After this, another layer of PMMA (495 K, A4, Microchem Company) was spin-coated on the substrate followed by a 5 min bake at 160 °C. Electron beam lithography (Raith 150 two) was employed to define the drain and source electrodes. After conventional development process, Ti/Au metal layer (Ti: 5 nm, Au: 50 nm) was deposited to form the source-drain electrodes by electron beam evaporation, and finally followed by lift-off process with acetone. The electrical and optoelectronic properties of the heterostructures were measured in vacuum ($10^{-6}$ torr) with the Lake Shore Probe Station and Keithley 4200 semiconductor analyzer at room temperature. The time response of the device was measured by switching the laser on and off with an internal square-wave trigger source and recorded by a digital oscilloscope.

**Data availability**. The authors declare that all of the data supporting the findings of this study are available within the article and its Supplementary Information file.

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

## Acknowledgements

This work was supported by the NSF of China (Nos. 51525202, 21521063, 61574054, 61505051, 61625402, 61474040, 61725505, and 11734016), the Hunan province science and technology plan (Nos. 2014FJ2001 and 2014TT1004), National Key Basic Research Program of China 2015CB921600, the Aid program for Science and Technology Innovative Research Team in Higher Educational Institutions of Hunan Province and the Fundamental Research Funds for the Central Universities.

## Author contributions

A.P., T.Y., and B.Z. conceived and designed the experiments. B.Z. synthesized the heterostructures. Y.F. and J.Z. performed the band structure calculation. T.Y., B.Z. carried out AFM and photoluminescence measurements. B.Z., T.Y., T.X., and L.S. carried out TEM measurements. T.Y., Z.W., and C.P. carried out the device fabrication and measurements. A.P., T.Y., and B.Z. wrote the paper with significant inputs from X.Z., Z.Q., H.L., W.H., F.M., and X.D. And A.P. supervised the research. All the authors participated in the analysis of the data and discussed the results. All the authors have read and approved the manuscript.

## Additional information

**Competing interests:** The authors declare no competing financial interests.

