## [Peer Review File · Nature Communications]

Reviewers' comments:

Reviewer #1 (Remarks to the Author):

The manuscript "Van der Waals epitaxial growth and optoelectronics of large-scale WSe₂/SnS₂ vertical bilayer p-n junctions" demonstrated the van der Waals epitaxy growth of WSe₂/SnS₂ vertical p-n junction for high performance devices. The advantage of WSe₂/SnS₂ p-n junction was systematically investigated through theoretical simulations, TEM, AFM and PL characterization, and was further demonstrated by FETs, photovoltaic and photodetectors measurements. The operation mechanisms of three kinds of devices were comparatively investigated. The results are impressive and breakthrough, especially with the controllable growth of the high-quality mm-size p-n junctions and the high performance of the devices, marking a vitally important step forward their future application in 2D integrated electronic and optoelectronic systems. The manuscript is well organized and written. Based on the enough originality of the work, I strongly recommend it to be published in the high impact Nature Communications after addressing the following issues.

1. In figures 1d and e, it seems that the morphology of the achieved WSe₂/SnS₂ heterostructure flake is usually triangle in your work, can you get any other shapes ?
2. As we know, the properties of two dimensional TMDs are related to the thickness. The band alignment of monolayer/monolayer WSe₂/SnS₂ is type-III, whether there is any difference when the thickness increases ?
3. In figure 3d, the inseted image looks plicated, with the surface of the WSe₂/SnS₂ flake not smooth, what happened ?
4. In figure 4f, the transfer curves of parallel-mode device demonstrate a ambipolar behavior. Why the n-type behavior looks so weak ? And the hole mobility of the parallel mode device is highest among these three kinds of channels, why the leakage current is larger, and the on-off ratio is smaller ?
5. In the work, the millimeter scale heterostructures are of great potential for practical applications. Have you considered any integrated systems based on the large scale 2D structures ?

Reviewer #2 (Remarks to the Author):

Yang et al reported a two-step CVD method to synthesize the WSe₂/SnS₂ 2D heterostructure and characterized with various techniques. The synthesized 2D heterostructure demonstrate the photoresponse properties. The authors didn't give the good review of the progress of 2D heterostructures in the introduction part. The very short general introduction doesn't explain why they select WSe₂/SnS₂ heterostrcutre. Actually, the growth of WSe₂/SnS₂ heterostructure is not new, which has been reported by Prof Song Jin at Nano Letter and Prof Zhongming Wei (2D Materials, 4, 025097). They didn't

cite any previous works on the growth of this heterostructure. The reported synthesis and characterizations of this heterostructure in this work are not novel so that I recommend to submit it to other special journal.

Reviewer #3 (Remarks to the Author):

Dear Editor,

the manuscript from Yang et al. deals with the complete research activity on a FET device based on vertical bilayer WSe₂/SnS₂ Van der Waals heterostructure. They grow the material, process a device on top of the bilayer and then characterize it using both charge transport and photocurrent measurements. I must say that the paper is very well written, extremely clear and the scientific conclusions sound and well supported by the experimental data. Therefore, I recommend to publish the manuscript in Nature Communications.

Nevertheless, I think I can help the authors by indicating some minor cosmetic changes which will definitely improve the paper.

- At first, I would recommend to increase the figures size and fonts, even at the cost of splitting some results in different figures. Especially the axis labels of figures 4 and 5 are very difficult to read.
- It is impressive that the authors can grow such large scale heterostructures. From my point of view (i.e. not an expert in CVD growth) it would be extremely interesting to know the yield of the good material obtained. Are the results showed in the paper more or less the same for every sample ? Or is it one beautiful device out of ten/hundreds ?
- I understand more or less the reasons for the increased mobility of the heterostructure with respect to the pristine material, but I feel that the physical discussion (lines 175 - 182) should be expanded a little or at least add some citations.
- I have found a typo at line 209 (everreported) and 211 (slot noise -> shot noise)

Point to point replies to the comments:

Comments from Reviewer #1:

The manuscript "Van der Waals epitaxial growth and optoelectronics of large-scale WSe₂/SnS₂ vertical bilayer p-n junctions" demonstrated the van der Waals epitaxy growth of WSe₂/SnS₂ vertical p-n junction for high performance devices. The advantage of WSe₂/SnS₂ p-n junction was systematically investigated through theoretical simulations, TEM, AFM and PL characterization, and was further demonstrated by FETs, photovoltaic and photodetectors measurements. The operation mechanisms of three kinds of devices were comparatively investigated. The results are impressive and breakthrough, especially with the controllable growth of the high-quality mm-size p-n junctions and the high performance of the devices, marking a vitally important step forward their future application in 2D integrated electronic and optoelectronic systems. The manuscript is well organized and written. Based on the enough originality of the work, I strongly recommend it to be published in the high impact Nature Communications after addressing the following issues.

Reply: We greatly appreciate that the reviewer was full of praise and gave a very positive comment to our work, and recommend it to be published in the high impact Nature Communications.

Question 1. *In figures 1d and e, it seems that the morphology of the achieved WSe₂/SnS₂ heterostructure flake is usually triangle in your work, can you get any other shapes ?*

Reply: Yes, we can also find some hexagon shape WSe₂/SnS₂ heterostructures in the sample, but most of the obtained heterostructures are triangle like shape, as showed in the manuscript.

Question 2. *As we know, the properties of two dimensional TMDs are related to the*

thickness. The band alignment of monolayer/monolayer WSe₂/SnS₂ is type-III, whether there is any difference when the thickness increases ?

Reply: Yes, the band alignment of WSe₂/SnS₂ heterostructure is dependent on its thickness. Our preliminary band structure calculation results indicate that the band alignment is type-III for the monolayer/monolayer structure as investigated in the current work, and it will be changed to type-II when the thickness is increased to few or more layers.

Question 3. *In figure 3d, the inseted image looks plicated, with the surface of the WSe₂/SnS₂ flake not smooth, what happened ?*

Reply: Because the very large lateral size of our WSe₂/SnS₂ heterostructure, we need use the stitch function of the microscope (WITec alpha-300) to get a clear optical image of the full examined flake, which make the surface of the sample look not very smooth.

Question 4. *In figure 4f, the transfer curves of parallel-mode device demonstrate a ambipolar behavior. Why the n-type behavior looks so weak ? And the hole mobility of the parallel mode device is highest among these three kinds of channels, why the leakage current is larger, and the on-off ratio is smaller ?*

Reply: For the parallel-mode device, the WSe₂ layer lies in the bottom, close to the back gate dielectric layer (SiO₂), while the SnS₂ layer lies in the top position, away from the SiO₂ layer, which induce that the gate control efficiency of SnS₂ is a lot lower than that of WSe₂, and finally cause the weak n-type behavior. For the same reason, the carries in the parallel-mode device are hard to be depleted due to the low gate control efficiency, leading to a high leakage current and a small on-off ratio.

Question 5. *In the work, the millimeter scale heterostructures are of great potential*

for practical applications. Have you considered any integrated systems based on the large scale 2D structures ?

Reply: Yes, we also planed to realize some integrated electronic systems, like CMOS, logic “AND” and “NOR” gates using these large scale p-n junctions. The related works are in progress.

Comments from Reviewer #2:

Yang et al reported a two-step CVD method to synthesize the WSe₂/SnS₂ 2D heterostructure and characterized with various techniques. The synthesized 2D heterostructure demonstrate the photoresponse properties. The authors didn't give the good review of the progress of 2D heterostructures in the introduction part. The very short general introduction doesn't explain why they select WSe₂/SnS₂ heterostrcutre. Actually, the growth of WSe₂/SnS₂ heterostructure is not new, which has been reported by Prof Song Jin at Nano Letter and Prof Zhongming Wei (2D Materials, 4, 025097). They didn't cite any previous works on the growth of this heterostructure. The reported synthesis and characterizations of this heterostructure in this work are not novel so that I recommend to submit it to other special journal.

Reply: Thank you for your review and providing with the precious advice.

The whole story of our work mainly focuses on the controlled vapor growth of large-scale atomic layered 2D p-n heterostructures, which is always a great challenge in the research field of 2D materials and technologies, for their further applications in integrated devices and systems.

Actually, the work published in Nano Letters by Prof Song Jin reported the growth of micrometer scale sized heterostructures, with few layers of MoS₂, WS₂ and WSe₂ grown on microplates of SnS₂. Both the thickness (micrometer scale) and the lateral size (up to ~30 μm) are far from our case. Moreover, their heterostructures were not Van der Waals epitally grown on the substrate, resulting in the random orientation of

their heterostructures, i.e. the flakes lie on the substrate in a random angle, which will cause unfavorable difficulties for the practical device applications. Also, they did not conduct any electrical transport and optoelectronic characterizations in this work. In all, the exhibited structures and research contents in the above work are both large different from ours in the current manuscript.

As for the work published in 2D Materials by Prof Zhongming Wei, though the material system of the heterostructures they investigated are the same as ours ($\text{WSe}_2/\text{SnS}_2$), the method they used is mechanical exfoliation and PMMA-assisted transferring, which is totally different to our direct vapor growth. Furthermore, the heterostructures they showed is not atomic layered, and their lateral sizes are only ~ 40 μm , which is far more smaller than ours. The story we want to tell in the current work is the achievement of large scale 2D monolayer/monolayer heterostructures through direct vapor growth route, which is also different with what shown in the above work.

Based on the big differences of our work with the above two works mentioned by the reviewer, we did not specially introduced and cited them in the previous manuscript. However, after carefully examining the reviewer's comments, we think that **the reviewer gave very good advice that we should give more review of the progress of 2D heterostructures in the introduction part, especially need to add the citations of the two works mentioned by the reviewer**, since they used the same material systems with ours. In the revised manuscript, we have improved the writing of the introduction part, and explained why we selected $\text{WSe}_2/\text{SnS}_2$ heterostructures (see page 3, Lines 48-57 of the revised manuscript).

From the above discussion and comparison, our work realized a breakthrough on the controlled preparation of large scale atomic layered p-n heterostructures through a direct vapor growth route, paving the way towards nanoscale integrated device and system application. Based on this breakthrough of this work and the improvement according to the reviewer's advice, we think it reaches the level to be published in Nature Communications.

Comments from Reviewer #3:

The manuscript from Yang et al. deals with the complete research activity on a FET device based on vertical bilayer WSe₂/SnS₂ Van der Waals heterostructure. They grow the material, process a device on top of the bilayer and then characterize it using both charge transport and photocurrent measurements. I must say that the paper is very well written, extremely clear and the scientific conclusions sound and well supported by the experimental data. Therefore, I recommend to publish the manuscript in Nature Communications.

Nevertheless, I think I can help the authors by indicating some minor cosmetic changes which will definitively improve the paper.

Reply: We greatly appreciate that the reviewer was full of praise and gave a very positive comment to our work, and recommend it to be published in the high impact Nature Communications.

Question 1. *At first, I would recommend to increase the figures size and fonts, even at the cost of splitting some results in different figures. Especially the axis labels of figures 4 and 5 are very difficult to read.*

Reply: Thank you for your precious advice, we have increased the figures size and fonts in the new revised manuscript version (see pages 21 and 22 of the revised manuscript).

Figure 4.

Figure 5.

Question 2. *It is impressive that the authors can grow such large scale heterostructures. From my point of view (i.e. not an expert in CVD growth) it would be extremely interesting to know the yield of the good material obtained. Are the results showed in the paper more or less the same for every sample ? Or is it one beautiful device out of ten/hundreds ?*

Reply: Our growth is well controllable and repeatable, the yield of our large scale WSe₂/SnS₂ p-n junction can reach ~ 90%. We have fabricated tens of devices and conducted systematical comparative investigations. We found the results of can be well repeated, and the main rules are almost the same for most of the devices we tested (such as the mobility of the parallel-mode device is the highest among these three kinds of devices, the on-off ratio and the photodetectivity of the parallel-series mode device is highest and so on).

Question 3. *I understand more or less the reasons for the increased mobility of the heterostructure with respect to the pristine material, but I feel that the physical discussion (lines 175 - 182) should be expanded a little or at least add some citations.*

Reply: Thank you for your advice, we have expanded the discussion part for the issue of increased mobility and added some citations. (see lines 190-196 of the revised manuscript). The citations added are as follows.

- 40 Li, B. *et al.* Direct Vapor Phase Growth and Optoelectronic Application of Large Band Offset SnS₂/MoS₂ Vertical Bilayer Heterostructures with High Lattice Mismatch. *Advanced Electronic Materials*. **2**, 1600298, (2016).
- 60 Zhang, W., Wang, Q., Chen, Y., Wang, Z. & Wee, A. T. S. Van der Waals stacked 2D layered materials for optoelectronics. *2D Materials*. **3**, 022001, (2016).
- 61 Cui, X. *et al.* Multi-terminal transport measurements of MoS₂ using a van der Waals heterostructure device platform. *Nat Nanotechnol*. **10**, 534-540, (2015).

Question 4. *I have found a typo at line 209 (everreported) and 211 (slot noise -> shot noise)*

Reply: Thank you for your carefully reading of our manuscript, I have correct these two typos in the new revised manuscript version at line 222 and line 225.

REVIEWERS' COMMENTS:

Reviewer #1 (Remarks to the Author):

This version has been well revised according to previous suggestion. Thus I think that it can be accepted in current form.

Reviewer #2 (Remarks to the Author):

Recommend to publish it since all the concerns of reviewers' are solved.

Point to point replies to the comments from reviewers:

Comments from Reviewer #1:

This version has been well revised according to previous suggestion. Thus I think that it can be accepted in current form.

Reply: We greatly appreciate that the reviewer gave a very positive comment to our work, and recommend it to be published in the high impact Nature Communications.

Comments from Reviewer #2:

Recommend to publish it since all the concerns of reviewers' are solved.

Reply: We greatly appreciate that the reviewer gave a very positive comment to our work, and recommend it to be published in the high impact Nature Communications.